# Implicit Offline Reinforcement Learning via Supervised Learning

## Abstract

Offline *Reinforcement Learning (RL) via Supervised Learning* is a simple and effective way to learn robotic skills from a dataset collected by policies of different expertise levels. It is as simple as supervised learning and Behavior Cloning (BC), but takes advantage of return information. On datasets collected by policies of similar expertise, implicit BC has been shown to match or outperform explicit BC. Despite the benefits of using implicit models to learn robotic skills via BC, offline *RL via Supervised Learning* algorithms have been limited to explicit models. We show how implicit models can leverage return information and match or outperform explicit algorithms to acquire robotic skills from fixed datasets. Furthermore, we show the close relationship between our implicit methods and other popular *RL via Supervised Learning* algorithms to provide a unified framework. Finally, we demonstrate the effectiveness of our method on high-dimension manipulation and locomotion tasks.

## 1 Introduction

Large datasets of varied interactions combined with Offline RL algorithms are a promising direction to learning robotic skills safely (Levine et al., 2020). A practical and straightforward approach to leveraging large and varied robotics datasets is to convert the RL problem into a supervised learning problem (Emmons et al., 2021; Chen et al., 2021). *RL via Supervised Learning* (RvS) algorithms can be seen as return-conditional, -filtered, or -weighted BC. Here, we use RvS to refer to policies where the action distribution is conditioned on a return defined by the user or generated by the agent.

*Reinforcement Learning via Supervised Learning* algorithms are as simple as BC ones, but since they take advantage of the return information, they can leverage sub-optimal interactions. In order to be effective for datasets collected by policies of different expertise levels, these algorithms must model multi-modal joint distributions, e.g., over the actions and the return. Previous RvS methods either discretized the variables and could use multinomial distributions (Chen et al., 2021) or ignored the multi-modality of the variables and used simple distributions such as a Gaussian distribution (Kumar et al., 2019). This paper shows that implicit models can be excellent for modeling multi-modal distributions without the need to discretize the return and action variables. In addition, implicit models have been shown to better extrapolate and model discontinuities than explicit models allowing them to capture complex robotic behaviors (Florence et al., 2021). Despite the advantages of using implicit models, RvS algorithms have been limited to explicit models.

In this work, we bridge the gap between implicit models and RvS, and propose the first implicit RvS (IRvS) algorithm. We show that our IRvS algorithm is closely related to other RvS algorithms but has the advantages of 1) not requiring the user or a second neural network to specify a target return, and 2) not requiring continuous outputs to be discretized to capture multi-modal distributions. Furthermore, we demonstrate the superiority of IRvS over RvS to interpolate, model discontinuity, and handle higher-dimension action spaces in an environment with linear dynamics. Furthermore, using the same objective as IRvS, we derive a novel RvS algorithm formulation where the target return is obtained by maximizing the exponential tilt of the return. Finally, we show that our method achieves state-of-the-art on the difficult ADROIT and FrankaKitchen manipulation tasks and is competitive with Offline RL and RvS algorithms on a suite of robotic locomotion environments.

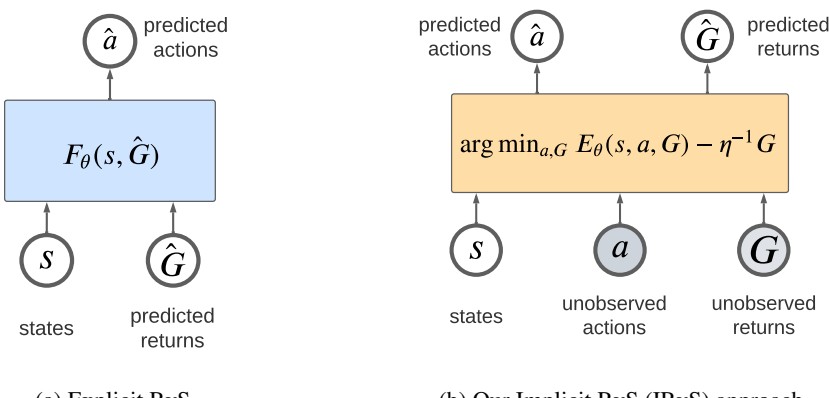

(a) Explicit RvS.      (b) Our Implicit RvS (IRvS) approach.

Figure 1: Comparison of implicit and explicit *Reinforcement Learning via Supervised Learning* (RvS). Implicit models can be more straightforward and use a single neural network. In contrast, explicit models require either a second neural networks (not depicted here) or the user to specify a task-specific target return $\hat{G}$.

## 2 PRELIMINARIES

**Offline Reinforcement Learning.** We consider a Markov Decision Process (MDP) $(\mathcal{S}, \mathcal{A}, \mathcal{P}, \mathcal{R})$ with states $\mathbf{s} \in \mathcal{S}$, actions $\mathbf{a} \in \mathcal{A}$, transition function $\mathcal{P}(\mathbf{s}'|\mathbf{s}, \mathbf{a})$, and reward function $r = \mathcal{R}(\mathbf{s}, \mathbf{a})$. We aim to learn a policy $\pi : \mathcal{S} \to \mathcal{A}$ maximizing the expected return $\mathbb{E}_{\tau \sim \pi}\big[G(\tau)\big]$, where $G(\tau) = \sum_{t=0}^{H} r_t$. Contrary to "online" RL agents that can interact with the environment, we focus on agents that must learn a policy exclusively from a dataset $\mathcal{D}$ of trajectories $\tau = (\mathbf{s}_0, \mathbf{a}_0, r_0, \ldots, \mathbf{s}_H, \mathbf{a}_H, r_H)$ collected under a behavior policy $\mu$. For the rest of the paper, we work with the tuple $(\mathbf{s}_i, \mathbf{a}_i, G_i)$, where $G_i = \sum_{t=i}^{H} r_i$ is the cumulative return starting from state-action $(\mathbf{s}_i, \mathbf{a}_i)$.

**Implicit Models and Energy-Based Models.** We define an *implicit model* as a general function approximator $E : \mathbb{R}^D \to \mathbb{R}$ where inference can be performed via optimization of $\hat{y} = \arg\min_{\mathbf{y}} E_\theta(x, y)$. In this work, we focus on *Energy Based Models* (LeCun et al., 2006) (EBMs). EBMs associate scalar energy to a configuration of variables $(\mathbf{x}, \mathbf{y})$. The model is trained to predict lower energy on the observed data than on the unobserved one. We use an EBMs to model the conditional density

$$p_{\boldsymbol{\theta}}(\mathbf{y}|\mathbf{x}) = \frac{\exp(-E_{\boldsymbol{\theta}}(\mathbf{x}, \mathbf{y}))}{Z_{\boldsymbol{\theta}}} \tag{1}$$

where $E_{\boldsymbol{\theta}}$ is the energy and the normalizing constant $Z_{\boldsymbol{\theta}} = \int_{\mathbf{y}} \exp(-E_{\boldsymbol{\theta}}(\mathbf{x}, \mathbf{y}))d\mathbf{y}$. Training the density model can be performed using the InfoNCE loss function (Oord et al., 2018) defined as

$$\mathcal{L}(\boldsymbol{\theta}) = \sum_{i=1}^{N} - \log(\tilde{p}_{\boldsymbol{\theta}}(\mathbf{y}_i \mid \mathbf{x}, \{\tilde{\mathbf{y}}_i^j\}_{j=1}^{N_{\text{neg.}}})) \tag{2}$$

$$\tilde{p}_{\boldsymbol{\theta}}(\mathbf{y}_i \mid \mathbf{x}, \{\tilde{\mathbf{y}}_i^j\}_{j=1}^{N_{\text{neg.}}}) = \frac{\exp(-E_{\boldsymbol{\theta}}(\mathbf{x}_i, \mathbf{y}_i))}{\exp(-E_{\boldsymbol{\theta}}(\mathbf{x}_i, \mathbf{y}_i)) + \sum_{j=1}^{N_{\text{neg.}}} \exp(-E_{\boldsymbol{\theta}}(\mathbf{x}_i, \tilde{\mathbf{y}}_i^j))}.$$

where $\{\tilde{\mathbf{y}}_i^j\}_{j=1}^{N_{\text{neg.}}}$ is a set of negative counter-examples. Negative sampling can be performed by minimizing $\tilde{\mathbf{y}}_i^j = \arg\min_{\mathbf{y}} E_{\boldsymbol{\theta}}(\mathbf{x}_i, \mathbf{y})$ using stochastic gradient Langevin dynamics (SGLD) (Welling & Teh, 2011), see Algorithm 1 for details. Finally, prediction can be done by minimizing the energy

$$\hat{\mathbf{y}} = \arg\min_{\mathbf{y} \in \mathcal{Y}} E_{\boldsymbol{\theta}}(\mathbf{x}, \mathbf{y}) \tag{3}$$

which can also be done via SGLD.

**Implicit Behavior Cloning.** Behavior Cloning (Pomerleau, 1988) is a simple and popular way to acquire robotic skills from a dataset of expert demonstrations. The policy is usually modeled via

an explicit model and learned by minimizing the mean squared error loss $\mathbb{E}_{\mathbf{s},\mathbf{a}\sim\mathcal{D}}\left[(\mathbf{a} - f_{\boldsymbol{\theta}}(\mathbf{s}))^2\right]$ to clone the behavior policy $\mu$. It is particularly effective when the dataset consists of expert trajectories. Florence et al. (2021) introduced *Implicit Behavior Cloning* where they leverage an EBM to learn a policy from offline data. They showed that EBMs could capture multi-modal distribution, interpolate, and model discontinuities making them particularly effective on continuous control tasks. The EBM is trained to learn the conditional density

$$p_{\boldsymbol{\theta}}(\mathbf{a}|\mathbf{s}) = \frac{\exp(-E_{\boldsymbol{\theta}}(\mathbf{s},\mathbf{a}))}{Z_{\boldsymbol{\theta}}}. \tag{4}$$

The decision-making is performed by minimizing the energy

$$\hat{\mathbf{a}} = \arg\min_{\mathbf{a}\in\mathcal{A}} E_{\boldsymbol{\theta}}(\mathbf{s},\mathbf{a}). \tag{5}$$

which can be minimized using SGLD. The advantages of EBMs allowed IBC to outperform explicit methods by 100% in some environments. While IBC performs particularly well on expert data, it does not leverage return information, enabling it to leverage more diverse non-expert data. This is an important limitation, as sub-optimal data might be easier to obtain, have better state coverage and result in better representation than expert data.

## 3 IMPLICIT OFFLINE RL VIA SUPERVISED LEARNING

### 3.1 METHOD

Inspired by the successes of EBMs on robotic tasks, we introduce *Implicit Reinforcement Learning via Supervised Learning* (IRvS) to improve on the IBC algorithm by modeling the dependencies between the state, action, and return variables. Modeling these dependencies would allow the robotics community to use implicit methods on varied datasets instead of being limited to datasets of expert demonstrations. Specifically, we model the following state conditional distribution

$$p_{\boldsymbol{\theta}}(\mathbf{a}, G|\mathbf{s}) = \frac{\exp(-E_{\boldsymbol{\theta}}(\mathbf{s},\mathbf{a},G))}{Z_{\boldsymbol{\theta}}}. \tag{6}$$

We now have access to a differentiable joint density model $p_{\boldsymbol{\theta}}(\mathbf{a}, G \mid \mathbf{s})$ from which we can sample action and return pairs using SGLD. Doing so results in actions likely under the training dataset irrespective of their associated returns, which is equivalent to the IBC algorithm. Instead, we want to bias sampling towards actions with high returns.

To sample action with high return, we instead sample from the *exponential tilt* (Asmussen & Glynn, 2007) in $G$ of the state conditional distribution. Specifically, we use the exponential tilt in $G$ defined as

$$p_{\boldsymbol{\theta}}(\mathbf{a}, G \mid \mathbf{s}_t; \eta) = p_{\boldsymbol{\theta}}(\mathbf{a}, G \mid \mathbf{s}_t)\exp(\eta^{-1}G - \kappa(\eta)), \tag{7}$$

where $\kappa(\eta)$ is the *cumulant generating function* $\log\mathbb{E}[e^{\eta^{-1}G}]$ and $\eta$ controls the trade-off between the likeliness of an action-return pair under the model and the magnitude of the return. We note that the exponential tilt density is still differentiable and can be efficiently sampled via SGLD. Specifically, the policy (and its associated return) can be obtained by maximizing the log density of the exponential tilt

$$\hat{\mathbf{a}}, \hat{G} = \arg\min_{\mathbf{a}\in\mathcal{A}, G\in\mathcal{G}} E_{\boldsymbol{\theta}}(\mathbf{s},\mathbf{a},G) - \eta^{-1}G, \tag{8}$$

where we can safely ignore the normalizing constant $\kappa(\eta)$. Informally, the above scheme generates high, but plausible returns under the model. For the rest of the paper, we omit $\kappa(\eta)$ to allege the notation.

### 3.2 DIDACTIC EXAMPLE

We build a simple didactic example to better understand how the hyper-parameter $\eta^{-1}$ changes the behavior of the IRvS algorithm. The environment has linear dynamics and goals located at each corner of the room. Each goal yields a different reward (-1, -1/4, 1/2, 1), and the dataset is collected

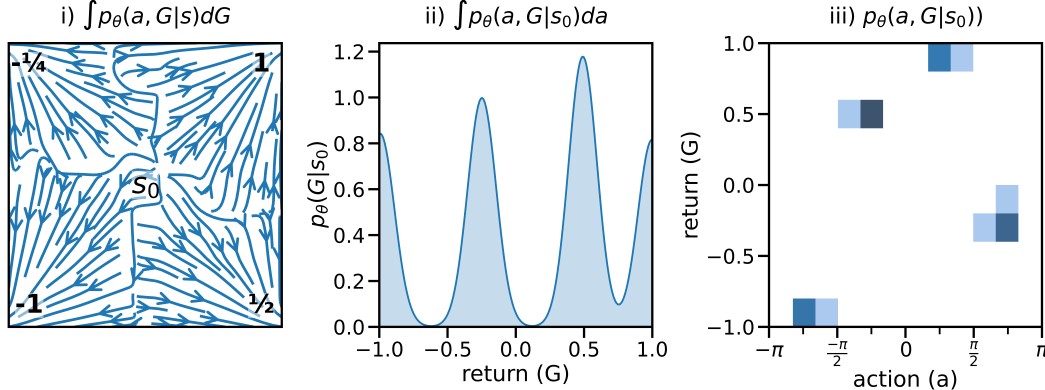

(a) **IRvS** $\eta^{-1} = 0$. In Figure i), we observe that the policy is the same as the goal agnostic policy that collected the data. In Figure ii), we observe that IRvS can successfully model the 4 possible non-discounted returns. In Figure iii), we observe that IRvS can successfully capture the joint distribution over actions and returns and their dependencies, where $a = \arctan \frac{\Delta y}{\Delta x}$.

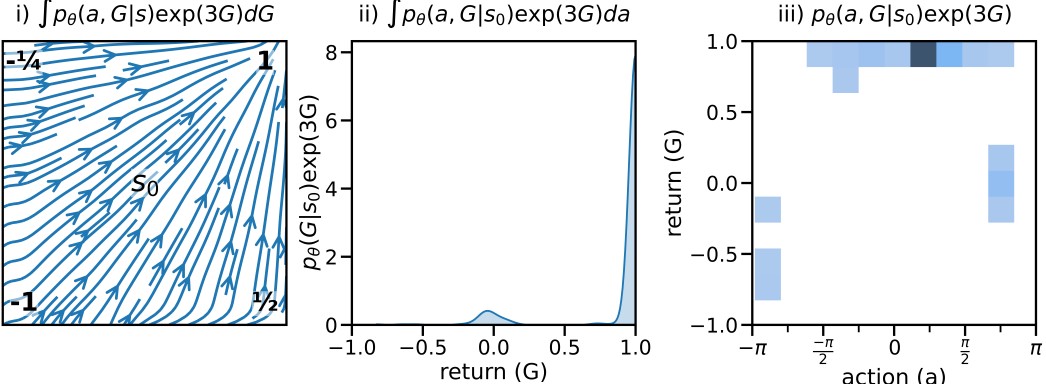

(b) **IRvS** $\eta^{-1} = 5.0$. In Figure i), we observe that IRvS policy is now leading towards the largest goal for most of the state space. In Figure ii), we observe that the exponential tilt of the return distribution is biased towards the highest reward. In Figure iii), we observe that the mass is concentrated over the return of 1 and actions near $\pi/4$ which are the optimal actions and returns.

Figure 2: **Effect of** $\eta^{-1}$. This experiment demonstrates how IRvS can leverage sub-optimal demonstrations to achieve near-optimal behaviors by increasing the parameter $\eta^{-1}$. The change of behavior caused by increasing $\eta^{-1}$ from 0 to 3 can be observed in the policy, the return prediction, and their joint distribution.

under a behavior policy that visits each goal with equal probability. The dataset thus consists of 25% of optimal demonstrations and 75% sub-optimal ones. The 2d actions $(\Delta_x, \Delta_y)$ are represented as $a = \arctan \frac{\Delta y}{\Delta x}$.

In Figure 2a, we study the behavior of IRvS when $\eta^{-1} = 0$ and is equivalent to IBC. Since the resulting policy is reward-agnostic, we expect the policy learned by IRvS to visit each goal with equal probability which can indeed be observed in the first sub-figure. The IRvS algorithm also successfully models the distribution of non-discounted returns which are possible to reach from the starting state. We note that IRvS can model the continuous return distribution, while RvS would need the returns to be discretized to capture the multi-modality of the distribution, see Figure 3. Finally, we observe that the joint distribution over returns and actions is successfully captured in the third figure.

In Figure 2b, we study the impact of increasing $\eta^{-1}$ to 3. In the first sub-figure, we observe that the policy is now heading towards the largest reward for most of the state space. In the second sub-figure, the distribution mass over return has now mostly shifted towards larger returns which are

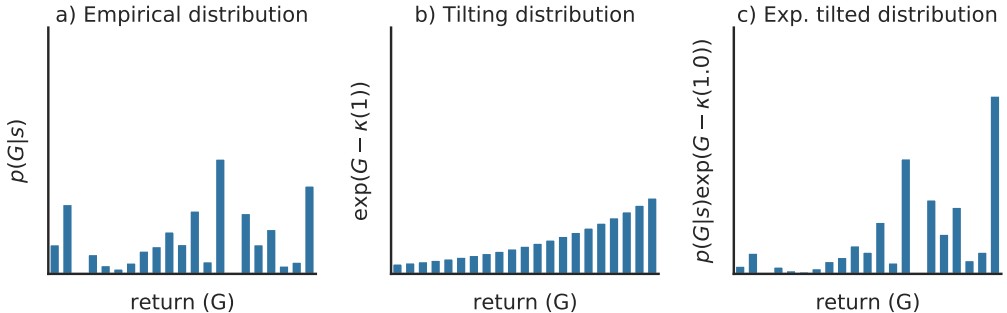

Figure 3: For RvS, we can use the exponential tilt of a learned distribution over the return to obtain a distribution biased towards higher returns. As we increase $\eta^{-1}$, the mass shifts towards higher returns. A limitation of these RvS methods is that the continuous distribution of return has to be discretized as shown here. While our proposed implicit method does not require the return to be discretized.

likely under the dataset, i.e., +1. In the third sub-figure, we observe that the most likely action is $\pi/4$ which is pointing towards the best goal.

## 4 RELATED WORK

### 4.1 IMPLICIT MODELS AND REINFORCEMENT LEARNING

We note that other works have used EBMs in RL. Interestingly, most previous works treat the $Q$-value as the energy and do not model how likely the $Q$-value is (or the return in our case). Sallans & Hinton (2004) parametrize the Restricted Boltzmann Machine (Freund & Haussler, 1991; Welling et al., 2004) negative energy function as the Q value. And more recently, Haarnoja et al. (2017) learn a $Q$ function via the (soft-) Bellman equation, then treat the $Q$ function as the negative energy function, which is sampled via a stochastic neural network. Recently, EBM techniques have been used to learn conservative $Q$-values (Kostrikov et al., 2021).

### 4.2 CONNECTION TO RETURN CONDITIONAL POLICIES

Offline *RL via supervised learning* is often done by learning a return conditional policy (Srivastava et al., 2019; Kumar et al., 2019; Emmons et al., 2021; Chen et al., 2021). Specifically, the policy $p(\mathbf{a}|\mathbf{s}, G)$ is conditioned on the return $G$, which can easily be learned via supervised learning and does not require *Temporal-Difference learning* (Sutton, 1988). At test time, the policy requires a return-to-go to be conditioned on. It is often treated as a task-dependent hyper-parameter (Chen et al., 2021; Emmons et al., 2021) or as a biased estimate of the return (a scalar instead of a distribution) estimated by a second neural network (Kumar et al., 2019).

Inspired by the exponential tilt formulation used for IRvS, we propose instead to model the state conditional distribution over returns $p(G \mid \mathbf{s})$ using supervised learning by discretizing the returns, see Figure 3. Similarly to IRvS, the return to go can be obtained by maximizing the exponential tilt

$$\hat{G}_t = \arg\max_G p(G \mid \mathbf{s}_t) \exp(\eta^{-1}G - \kappa(\eta)) \tag{9}$$

resulting in a high return which is also likely under the learned model. The three distributions are depicted in Figure 3. Thus, we argue that if the action is obtained by maximizing the learned likelihood of the model, e.g., outputting the mean of a Gaussian distribution,

$$\hat{\mathbf{a}} = \arg\max_{\mathbf{a}} p_{\boldsymbol{\theta}}(\mathbf{a} \mid \mathbf{s}_t, \hat{G}_t). \tag{10}$$

Thus, RvS and IRvS obtain the actions by maximizing the same objective, and their main differences are in the modeling choices. First, using an implicit model is simpler than an explicit one since it

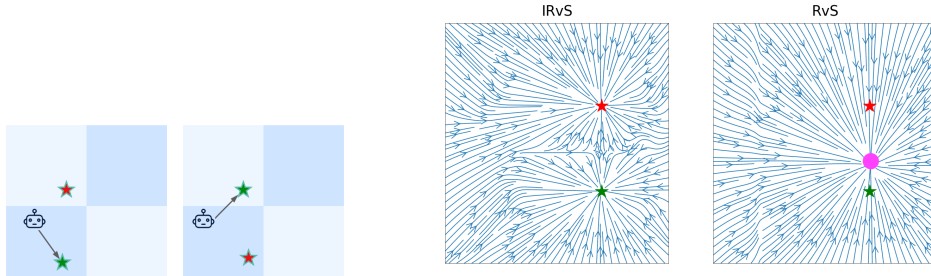

(a) The agent must reach the left-most goal. (b) Near discontinuities, IRvS reaches both goals, while RvS fails to reach either goal and converges to the purple point.

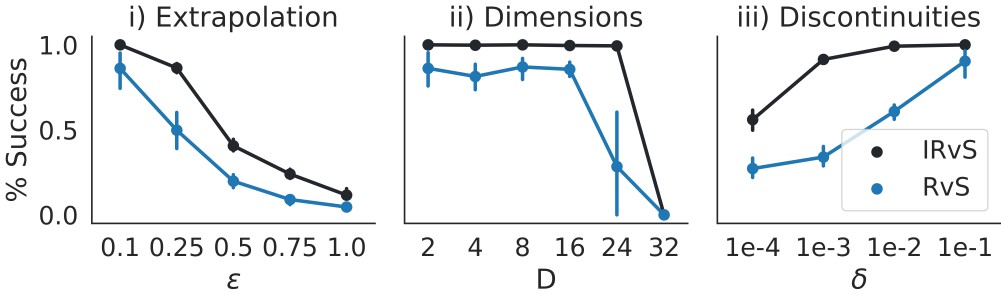

(c) Difference in performance between implicit RvS and (explicit) RvS.

Figure 4: **Comparison between Implicit and Explicit RvS method.** Figure a) shows how the agent changes focus from one goal to the other depending on which one is placed more to the left. Figure b) depicts how implicit vs explicit RvS behave when the goal to reach is unclear. Figure c) measure the performance of implicit and explicit RvS method along extrapolation capabilities, environment dimensionality, and policy discontinuities.

requires only a single neural network, $p_{\boldsymbol{\theta}}(\mathbf{a}, G|\mathbf{s})$, compared to two networks for an explicit model, i.e., $p_{\boldsymbol{\theta}}(\mathbf{a} \mid \mathbf{s}_t, \hat{G}_t)$ and $p_{\boldsymbol{\theta}}(\hat{G}_t \mid \mathbf{s}_t)$, see Figure 1. Second, implicit models can model rich continuous distributions making them a natural tool for control tasks, while explicit models must discretize continuous variables or assume a parametric distribution such as the Gaussian to model the multi-modal action and return distributions. In summary, by modeling the dependency between the state, action, and return with an implicit model, we bridge an important gap between the popular IBC and RvS methods.

Concurrently, Lee et al. (2022) introduced a binary "expert" variable, which leads to a similar way to select the return-to-go on, which the policy is conditioned on. They then demonstrated impressive performance using a Transformer on the Atari suite. In order to model the distribution over the return-to-go, they discretize the variable and model it with a categorical distribution. We note that introducing a binary "expert" variable is closely related to the concept of "optimality" in control-as-inference (Levine, 2018).

# 5 EXPERIMENTAL RESULTS

## 5.1 ADVANTAGES OF IMPLICIT RvS OVER EXPLICIT RvS

In order to understand the advantages of implicit RvS over explicit RvS, we proposed a simple navigation task with linear dynamics, where the agent has to reach the left-most goal, i.e. the goal with the smallest first dimension. The optimal policy while being relatively simple is *discontinuous* as the left-most goal changes. For example, in Figure 4a, slightly moving the bottom goal left results in a completely different optimal action.

**Experimental settings.** For all experiments, we keep the number of demonstrations constant at 100,000. The agent's initial position $\mathbf{s}$ is sampled uniformly from $[-1, 1]^D$. The 2 goals during training are independently sampled uniformly from $[-0.1, 0.1]^D$. The state is composed of the agent's position and two goals, resulting in a state of dimensions $3 \times D$, The optimal action is $\mathbf{a}^* = g^* - \mathbf{s}$ which is also of dimension $D$, where $g^*$ is the left-most goal. To better understand the advantages of implicit models over explicit ones, we measure i) the performance as we increase the dimensionality of the environment, ii) the ability to model discontinuous policies, and iii) the generalization capabilities of the model as we change the goal sampling distribution.

**Extrapolation.** Implicit models have been shown to extrapolate more effectively than explicit ones. This environment demonstrates this is also the case for implicit versus explicit RvS. To study extrapolation, the dataset is composed of demonstrations where the goals are placed near the origin, i.e., $g$ is sampled on $[-0.1, 0.1]^D$. At the same time, we can increase the goal sampling radius ($\epsilon$) at test time to measure the extrapolation capabilities of the model. In Figure 4c a) we observe that implicit RvS extrapolates more effectively than explicit RvS to changes in goal radius much larger than observed during training.

**High-dimensions.** As we increase the environment's dimensionality but keep the dataset's size constant, the training demonstrations are, on average, further away from what is observed at test time. Thus increasing the dimensions of the environment increases the difficulty of the task. In Figure 4c b), we can observe that implicit RvS performance degrades more gracefully than explicit RvS. Similar behavior has also been observed in Florence et al. (2021).

**Discontinuities.** Looking at Figure 4a, we can see that as the bottom goal slide to the right, the optimal action switches from going to the bottom goal to the top one: making the optimal policy highly discontinuous. This represents a challenge for explicit models as they can hardly fit discontinuous functions and instead fit intermediate values. We can measure the models ability to handle discontinuous functions by controlling the relative position of the second goal. Specifically, instead of sampling two goals independently (as done above), we can set the second goal's first dimension equal to the first goal's first dimension with a small amount of uniform noise sampled from $[-\delta, \delta]$. In Figure 4b, we observe qualitatively the behavior of the two models for $\delta$=1e-4. We observe that the (explicit) RvS policy converges to the purple point which is between the two goals and rarely reaches one of them, while IRvS policy reaches the correct goal half the time. Finally, in Figure 4c, we quantify the differences between the policies by measuring their success rate for different $\delta$. We see that IRvS outperforms RvS for all $\delta$.

## 5.2 CONTINUOUS CONTROL DATASET

We compare both IRvS and our version of RvS (as defined by Equation (9) and Equation (10)) to implicit and explicit state-of-the-art algorithms on a suite of offline RL tasks provided by d4rl (Fu et al., 2020). Specifically, we compare our methods to conservative Q-learning (Kumar et al., 2020), to the implicit algorithms IBC and IBC w/ RWR proposed by Florence et al. (2021), and to the explicit algorithms BC from Fu et al. (2020), and *RvS hand-tuned* (the target returns were selected by hand on a task-level) from Emmons et al. (2021).

### 5.2.1 EXPERIMENT SETTINGS.

We fine-tuned the hyper-parameter $\eta^{-1}$ for our algorithm on 3 seeds, see Appendix A.3. Then we run 3 seeds on the whole suite where we evaluate our algorithms for 20 episodes, for task-level results see Table 1. As in Emmons et al. (2021) we define $G$ as the average future reward. Additionally, we normalize each state coordinate to have mean 0 and standard deviation 1, and the actions and the return $G$ to lie in $[-1, 1]$.

**ADROIT.** The ADROIT benchmark (Kumar et al., 2013; Rajeswaran et al., 2017) consists of 4 environments (relocate, pen, door, hammer) and 3 datasets (human, cloned, expert) for a total of 12 tasks. The agent controls a 24-DoF anthropomorphic manipulator and must solve challenging and dynamic dexterous manipulation. The first environment relocation requires the agent to move a ball to a randomized target. Similarly, the pen environment requires the agent to reposition a pen to match the orientation of a randomized target. The door environment requires the agent to undo the latch and swing the door open. Finally, the hammer environment requires the agent to pick up a hammer and drive a nail into a board. The human dataset is comprised of 25 human demonstrations. The

| | | | IRvS (Ours) $\eta^{-1} = 1$ | IRvS (Ours) Best $\eta$ | BC | CQL | RvS Hand-tuned | IBC w/ RWR | RvS (Ours) $\eta^{-1} = 10$ | RvS (Ours) Best $\eta$ | IBC |
|---|---|---|---|---|---|---|---|---|---|---|---|
| adroit | cloned | door | 6.3 | **13.0** | -0.1 | 0.4 | - | 0.1 | 0.1 | 0.1 | -0.1 |
| | | hammer | **3.9** | **3.9** | 0.8 | 2.1 | - | 0.3 | 2.8 | 3.1 | 0.3 |
| | | pen | 76.7 | **90.2** | 56.9 | 39.2 | - | 81.1 | 80.3 | 81.4 | 64.5 |
| | | relocate | 0.1 | **0.7** | -0.1 | -0.1 | - | 0.1 | 0.0 | 0.1 | 0.0 |
| | expert | door | 103.7 | **105.8** | 34.9 | 101.5 | - | 104.4 | **105.4** | 105.5 | 101.7 |
| | | hammer | **127.2** | **127.2** | 125.6 | 86.7 | - | 118.0 | **127.0** | **127.0** | 123.4 |
| | | pen | 128.1 | **138.1** | 85.1 | 107.0 | - | 123.0 | 127.5 | 127.5 | 116.2 |
| | | relocate | 102.5 | **107.5** | 101.3 | 95.0 | - | 106.3 | 106.5 | 106.7 | 102.3 |
| | human | door | 15.6 | **16.5** | 0.5 | 9.9 | - | 13.2 | 2.9 | 4.4 | 11.0 |
| | | hammer | 1.5 | 1.7 | 1.5 | **4.4** | - | 1.2 | 1.9 | 2.3 | 1.3 |
| | | pen | **84.3** | **84.3** | 34.4 | 37.5 | - | 56.7 | 62.9 | 76.6 | 71.4 |
| | | relocate | **0.3** | **0.3** | 0.0 | 0.2 | - | 0.1 | 0.0 | 0.1 | 0.1 |
| kitchen | complete | kitchen | 65.4 | **78.8** | 33.8 | 43.8 | 1.5 | 69.1 | 45.8 | 46.9 | 76.2 |
| | mixed | kitchen | **67.9** | **67.9** | 47.5 | 51.0 | 1.1 | 52.1 | 46.5 | 47.9 | 39.6 |
| | partial | kitchen | 55.4 | **67.1** | 33.8 | 49.0 | 0.5 | 52.9 | 51.5 | 51.5 | 42.5 |
| mujoco | medium | halfcheetah | 41.7 | 42.1 | 36.1 | **49.1** | 41.6 | 41.6 | 42.8 | 42.8 | 41.5 |
| | | hopper | 61.7 | 61.7 | 29.0 | **64.6** | 60.2 | 54.1 | 59.4 | 59.4 | 53.3 |
| | | walker2d | 64.9 | 69.5 | 6.6 | **82.9** | 71.7 | 64.3 | 70.4 | 70.4 | 39.7 |
| | medium-expert | halfcheetah | **93.5** | **93.5** | 35.8 | 85.8 | 92.2 | 92.5 | 92.2 | 92.2 | 51.4 |
| | | hopper | 101.6 | 101.6 | **111.9** | 102.0 | 101.7 | 102.1 | 91.1 | 94.9 | 91.4 |
| | | walker2d | 107.2 | 107.2 | 6.4 | **109.5** | 106.0 | 105.8 | 107.8 | 108.3 | 97.8 |
| | medium-replay | halfcheetah | 34.4 | 34.4 | 38.4 | **47.3** | 38.0 | 37.4 | 40.0 | 40.0 | 26.6 |
| | | hopper | 25.0 | 27.2 | 11.8 | **97.8** | 73.5 | 17.2 | 61.7 | 63.0 | 1.8 |
| | | walker2d | 52.9 | 52.9 | 11.3 | **86.1** | 60.6 | 26.6 | 51.8 | 51.8 | 16.0 |

Table 1: **Normalized score per task.**. In general, IRvS performs well on high-dimensional manipulation tasks where the dataset has been collected by policies on varying expertise levels. While CQL and explicit methods perform well on the locomotion tasks. All algorithms perform well on the adroit expert dataset.

expert dataset is comprised of 5000 trajectories sampled from an expert policy. The cloned dataset is comprised of 2500 trajectories from the expert policy and 2500 trajectories from a behavior cloning policy trained on the human demonstrations.

**Mujoco.** The offline mujoco benchmark (Todorov et al., 2012) consists of 3 locomotion environments (halfcheetah, walker2d, and hopper) and 3 datasets (medium, medium-replay, medium-expert) for a total of 9 tasks. The hopper environment consists of a one-legged robot system with 11-dimensional state space and 3 degrees of freedom (DoF), halfcheetah is a cheetah robot with 11-dimensional state pace and 6-DoF, finally, walker2d has a 17-dimensional state space and 6-DoF. The medium datasets consist of $10^6$ samples, the medium-expert of $2 \times 10^6$ samples, and the medium replay of $10^5$ samples.

**FrankaKitchen.** The FrankaKitchen (Gupta et al., 2019) (kitchen) environment places a 9-DoF robot in a realistic kitchen environment where it can interact freely with different objects, e.g. opening the microwave and various cabinets, moving the kettle, and turning on the lights and burners. The task consists of reaching the desired configuration of the objects in the scene. The kitchen benchmark consists of 3 different tasks (complete, partial, and mixed). The complete dataset includes demonstrations of all 4 target sub-tasks being completed in order. The partial dataset includes consists of multiple other tasks being performed including the 4 target sub-tasks being completed in sequence. Finally, the mixed dataset consists of various sub-tasks, but the 4 target sub-tasks are never completed in sequence together.

### 5.2.2 RESULTS

**ADROIT.** On 11 of the 12 datasets, IRvS obtains or matches the best performance. We also note that, excluding IRvS, IRvS $\eta^{-1} = 1$ obtains the best performance on 6 of the 12 datasets. With the exception of Conservative Q-learning (CQL) (Kumar et al., 2020) obtaining the best score on hammer-human, implicit methods outperform explicit ones on most tasks. The gap between IRvS and RvS is particularly noticeable for the human and cloned datasets hinting that implicit methods perform better for high-dimensional control tasks combined with datasets collected by policies of different proficiency.

**Mujoco.** CQL obtains the best score on 8 of the 9 datasets while IRvS $\eta^{-1} = 1$ obtains the best score on halfcheetah medium-expert. These results suggest that TD learning and *trajectory stitching* are crucial on the mujoco benchmark to obtain the best performance. Interestingly, IRvS and *RvS*

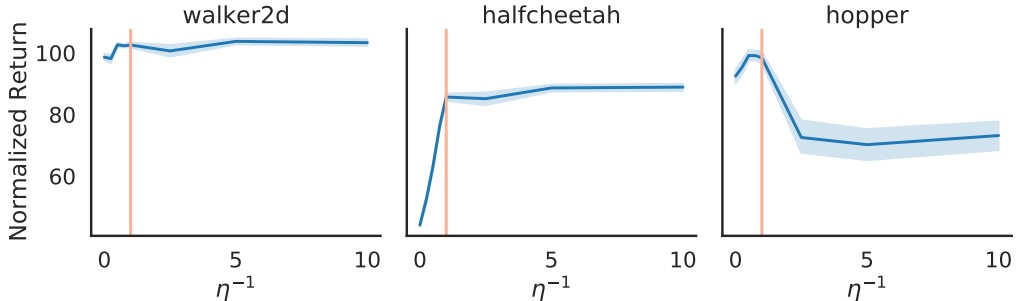

Figure 5: **Effectiveness of $\eta^{-1}$ to incorporate return information.** We report the performance of IRvS as a function of $\eta^{-1}$ on the medium-expert datasets. For walker2d, we observe that increasing $\eta^{-1}$ slightly improves the performance, but does not impact performance too much past a certain point. For halfcheetah, we see that increasing $\eta^{-1}$ dramatically improves the performance. For hopper, we observe that increasing $\eta^{-1}$ improves the performance up to a point, then the return falls below the behavior cloning performance ($\eta^{-1} = 0$). These observations confirm that $\eta^{-1}$ allows us to trade-off between the likelihood of a behavior and its potential return. We denote $\eta^{-1} = 1$ (the best overall hyper-parameter across all tasks) by red vertical lines.

*hand-tuned* (Emmons et al., 2021) obtain almost identical performance on the medium and medium-expert datasets, while RvS obtains much better scores on the medium-replay datasets. It is worth noting that our implementation of RvS $\eta^{-1} = 10$ achieves roughly the same score as *RvS hand-tuned* on most of the locomotion tasks, where *RvS hand-tuned* condition on the following hand-selected return: 110 for the medium-expert dataset, 90 for hopper, walker2d-medium-replay, and walker2d-medium, 60 for hopper-medium, and 40 for halfcheetah-medium, medium-replay.

**FrankaKitchen.** IRvS reaches the best score on all 3 datasets. Additionally, if excluding IRvS, IRvS $\eta^{-1} = 1$ reaches the best score on 2 of the 3 tasks, while IBC (Florence et al., 2021) obtains the best score on kitchen complete - which is well suited for behavior cloning. Surprisingly, RvS $\eta^{-1} = 10$ outperforms by a wide margin *RvS hand-tuned*. Overall, implicit methods outperform explicit methods for all kitchen tasks.

**Sensitivity Analysis.** In Figure 5, we observe that for every task the best performance is obtained with $\eta^{-1} > 0$. Thus confirming the effectiveness of $\eta^{-1}$ to incorporate return information by outperforming IBC ($\eta^{-1} = 0$). Interestingly, increasing $\eta^{-1}$ beyond a point does not impact the performance for walker2d. However, increasing $\eta^{-1}$ beyond a point decreases the performance on hopper, thus implying that some tasks cannot be solved by naively conditioning on the maximum observed return. Finally, while the best performance is reached using different $\eta^{-1}$, we found that using $\eta^{-1} = 1$ (red vertical line) for all tasks results in good overall performance. Additional figures on the sensitivity of $\eta^{-1}$ can be observed in Appendix A.3.

## 6 CONCLUSION

This work bridges an essential gap between implicit models and RvS methods. By bridging this gap, practitioners will be able to enjoy the advantages of implicit models to learn robotic skills from an offline dataset of varied expertise. Overall, IRvS is conceptually simpler than previous explicit RvS methods as it requires a single network and does not require quantizing the return. In addition to being simpler, we have shown that IRvS can better capture discontinuities, extrapolate, and handle higher dimensional problems than explicit RvS. Additionally, we use insight from IRvS to derive a simpler RvS algorithm where the target return is controlled via a temperature $\eta$ instead of being specified by hand or treated as a scalar. Furthermore, we have demonstrated performance matching or exceeding state-of-the-results on challenging offline RL benchmarks, particularly on the high-dimensional manipulation tasks where IRvS greatly outperforms the baselines. Finally, we have performed a sensitivity analysis where we have shown that our newly introduced hyper-parameter $\eta^{-1}$ can be set to 1 on all tasks without substantially degrading performance.

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

# A  APPENDIX

## A.1  Stochastic Gradient Langevin Dynamics

---

**Algorithm 1** Stochastic Gradient Langevin Dynamics

---

**Require:** Initial state $\mathbf{s}$
**Require:** Initial action $\mathbf{a} \sim \mathcal{U}(\mathbf{a}_{\min}, \mathbf{a}_{\max})$
**Require:** Initial return $G \sim \mathcal{U}(-1, 1)$
**Require:** Learning rate $\alpha$
1: *# Density Estimation*
2: **for** $i \in \{1, \ldots, 100\}$ **do**
3:     $g \leftarrow \nabla_{(\mathbf{a}, G)} E_\theta(\mathbf{s}, \mathbf{a}, G)$
4:     $u \leftarrow \text{clip}(\frac{1}{2}g + \epsilon, -0.5, 0.5), \epsilon \sim \mathcal{N}(0, 1)$
5:     $(\mathbf{a}, G) \leftarrow (\mathbf{a}, G) - \alpha u$
6: **end for**
7: **Return:** $(\mathbf{s}, \mathbf{a}, G)$

---

## A.2  Hyper-parameters

For IRvS, all the hyper-parameters with the exception of the learning rate are the same as the ones reported by Florence et al. (2021). For our implementation of RvS, we closely followed the hyper-parameters reported by Emmons et al. (2021).

| Hyper-parameter | Chosen Value | Swept Value |
|---|---|---|
| EBM variant | Langevin | |
| batch size | 512 | 512 |
| learning rate | 1e-3 | 1e-4, 5e-4, 1e-3 |
| learning rate decay | 0.99 | |
| learning rate decay steps | 100 | |
| network size (width x depth) | 512x8 | |
| activation | ReLU | swish, leaky ReLU, ReLU |
| regularization | none | none, layer norm |
| layer type | spectral norm | regular, spectral norm |
| train counter-examples | 8 | |
| action boundary buffer | 0.05 | |
| gradient penalty | final step only | |
| gradient margin | 1.0 | |
| Langevin iterations | 100 | |
| Langevin learning rate init. | 0.5 | |
| Langevin learning rate final | 1e-5 | |
| Langevin polynomial decay power | 2 | |
| Langevin delta action clip | 0.5 | |
| Langevin noise scale | 0.5 | |
| Langevin 2nd iteration learning rate | 1e-5 | |
| Didactic example gradient steps | 2000 | |
| Navigation task gradient steps | 10000 | |
| Continuous control gradient steps | 100000 | |

Table 2: Hyper parameter for IRvS

**CQL Results.** All the CQL results are from Fu et al. (2020), except for the mujoco results which are from Emmons et al. (2021), since the results were from the previous simulator version.

**IBC and filter-IBC.** We have reproduced the results from Florence et al. (2021) such that IBC, IRvS, and IBC w/ RWR share the same code-base.

**RvS.** We report the results RvS results from Emmons et al. (2021) as RvS, we note that they condition on return of 110 for the medium-expert dataset, 90 for hopper, walker2d-medium-replay, and

| Hyper-parameter | Chosen Value | Swept Value |
|---|---|---|
| batch size | 16384 | 16384 |
| learning rate | 1e-3 | 1e-4, 5e-4, 1e-3 |
| network size (width x depth) | 2048x2 | |
| activation | ReLU | swish, leaky ReLU, ReLU |
| Navigation task gradient steps | 10000 | |
| Continuous control gradient steps | 10000 | |
| Scales | 1e-5 | 1e-4, 1e-5, 1e-6 |
| Number of atoms | 101 | 21, 51, 101 |

Table 3: Hyper parameter for RvS

walker2d-medium, 60 for hopper-medium, and 40 for halfcheetah-medium, medium-replay. Additionally, we reported the RvS $\eta^{-1} = 10$, where the action selection and return prediction are performed as described in Equation (10) and Equation (9) respectively.

## A.3 SENSITIVY ANALYSIS

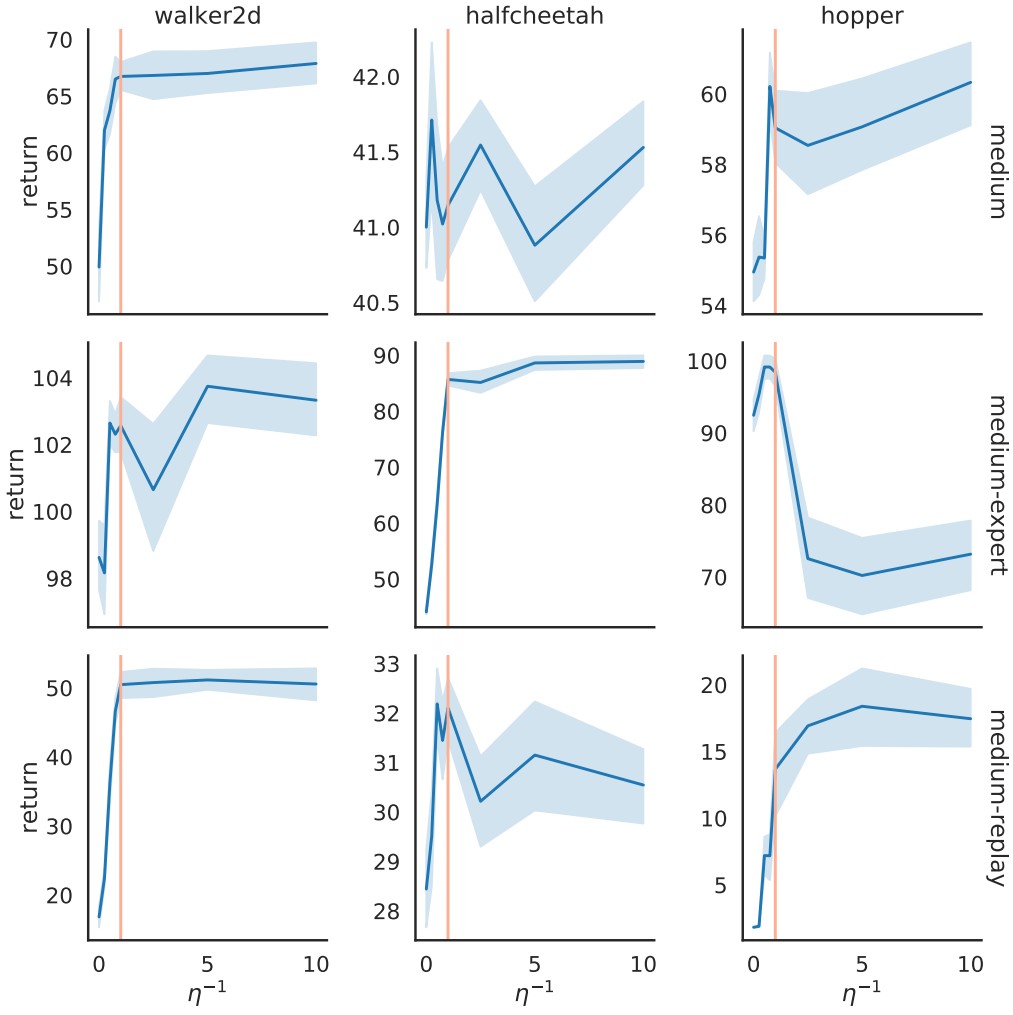

Figure 6: **Mujoco locomotion tasks.** Hyper-parameter selection was performed on these 3 seeds, and final results in Table 1 are reported on 3 new seeds.

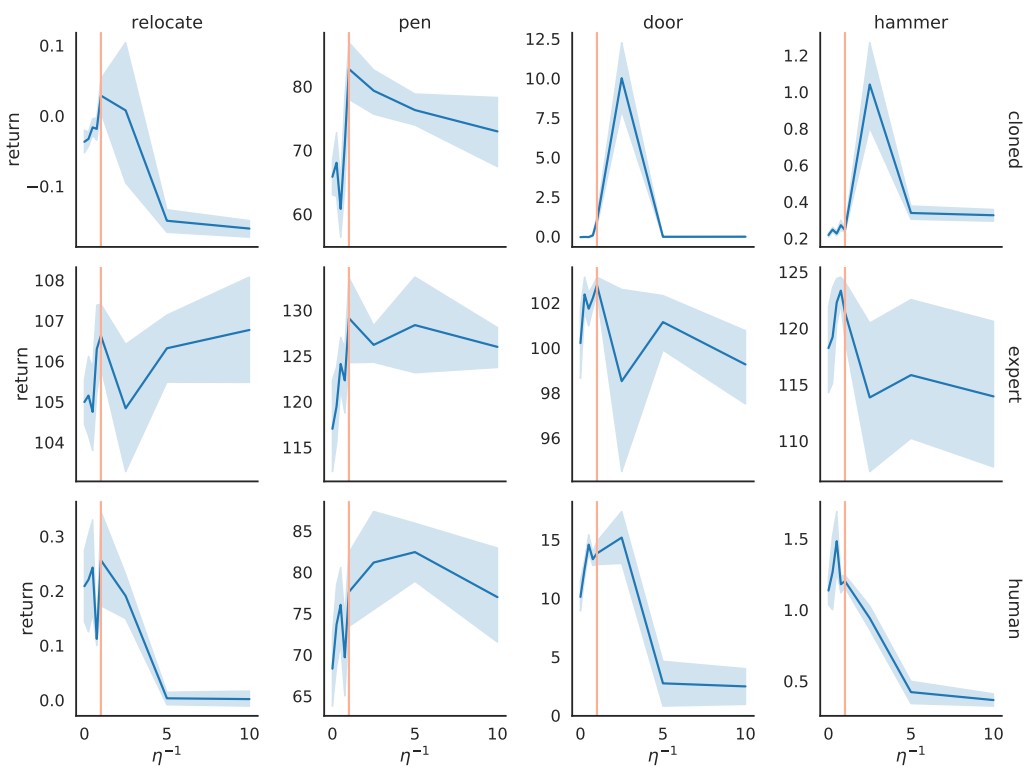

Figure 7: **Adroit manipulation tasks.** Hyper-parameter selection was performed on these 3 seeds, and final results in Table 1 are reported on 3 new seeds.

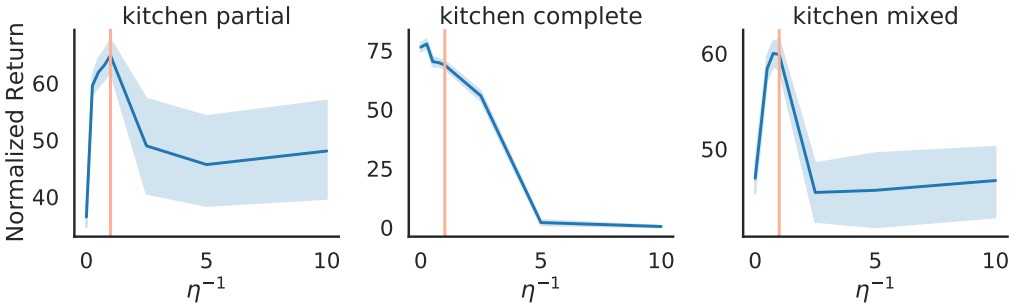

Figure 8: **Kitchen manipulation tasks.** Hyper-parameter selection was performed on these 3 seeds, and final results in Table 1 are reported on 3 new seeds.

