# OpenReview forum: "Implicit Offline Reinforcement Learning via Supervised Learning"
_ICLR.cc/2023/Conference — Submitted to ICLR 2023_

### Official Review · Reviewer_ExRr · 2022-10-17

**Confidence:** 4
**Correctness:** 3
**Technical Novelty And Significance:** 2
**Empirical Novelty And Significance:** 3
**Recommendation:** 5

**Clarity, Quality, Novelty And Reproducibility:**

The clarity and quality are good. The algorithm seems easy to reproduce. The proposed algorithm seems novel to me.

**Strength And Weaknesses:**

Strength:
The paper is well written and easy to follow. The idea is simple but proves to be effective in some domains.

Weakness:
1) The improvements in adroit and kitchen are impressive. However, the reason for the failure in Mujoco is not clear. Since the main contribution of this work is empirical, I feel the authors should dig into Mujoco more deeply to understand this failure.
2) I think some important baselines are missing, e.g., the authors should include comparison with offline RL methods with return-conditioned policy with explicit model, mentioned in Section 4.2.
3) Table 1 results from only 3 seeds without metrics on variation. So it is not as informative as it should be.

**Summary Of The Paper:**

The paper proposes to incorporate the return information in implicit behaviour cloning. Empirical study confirms the efficacy of the proposed algorithms in some domains

**Summary Of The Review:**

see above

---

> ### Author Response · Authors · 2022-11-09
> **Response to Reviewer ExRr**
>
>
> We want to thank the reviewer for taking the time to review our manuscript.
>
> > The improvements in adroit and kitchen are impressive.
>
> Thank you.
>
> > However, the reason for the failure in Mujoco is not clear. Since the main contribution of this work is empirical, I feel the authors should dig into Mujoco more deeply to understand this failure.
>
> We would not characterize our method as failing on mujoco. It is in fact competitive with RvS (i.e. a return-conditioned policy with an explicit model). Further gain in performance can also be achieved by using layer norm on the Mujoco tasks e.g., a return of ~50 can be achieved on Hopper medium-replay. We can add additional figures in the Appendix.
>
> > I think some important baselines are missing, e.g., the authors should include comparison with offline RL methods with return-conditioned policy with explicit model, mentioned in Section 4.2.
>
> We included 3 different baselines for RvS (methods based on a return-conditioned policy using an explicit model) e.g., RVS hand-tuned, RvS $\eta^{-1}=10$ and RvS best $\eta$. These are all the possible return-conditioned policy with explicit model based baselines that we can think of. Please let us know if you have something else in mind.
>
> > Table 1 results from only 3 seeds without metrics on variation. So it is not as informative as it should be.
>
> Please note that we have examined variation in figures 6, 7, and 8 of our Appendix. We are currently running 3 additional seeds (for a total of 6 seeds) and will provide the standard error in the main table.
>
> We hope that our responses to your comments and questions have addressed your concerns. If you have any other questions we are happy to provide further responses. We believe our work is of significant interest to the community, but we are considering withdrawing the paper, so it would be helpful if you could indicate your openness to increasing your score as a result of our responses, additional experiments in our appendix, and our proposed additional experimentation.

---

### Official Review · Reviewer_3Dga · 2022-10-23

**Confidence:** 4
**Correctness:** 3
**Technical Novelty And Significance:** 3
**Empirical Novelty And Significance:** 3
**Recommendation:** 5

**Clarity, Quality, Novelty And Reproducibility:**

Clarity: paper is well-written.
Quality: See strengths and weaknesses above. In the experiments, the proposed method does not convincingly work better than other approaches for robot tasks.
Novelty: Technical novelty is somewhat limited, as much of the proposed method does not introduce many novel concepts.
Reproducibility: The authors would need to provide better documented code, and with all environments, in order to reproduce this work.

**Strength And Weaknesses:**

Strengths:
- The paper is well-written and clear to follow.
- The proposed method is interesting and seems to be pretty simple and elegant.
- The toy examples and plots are nice in providing some intuition about the approach.

Weaknesses:
- It's not clear why IRvS is better than the naive way of incorporating return information by combining IBC with RWR. Furthermore, IRvS only does slightly better than IBC w/ RWR, and the performance is similar on most tasks. IRvS with the best eta does perform better, but requires tuning for eta. Much more thorough analysis is needed here, as this is the main contribution of the paper.
- Training EBMs with Langevin Dynamics can be difficult and requires many hyperparameters.
- Sometimes reward information is unavailable or difficult to get, particularly in real robotics, and the method may struggle in these cases.


**Summary Of The Paper:**

This paper proposes using implicit models for offline reinforcement learning. More specifically, this paper proposes using an implicit model to model the distribution of actions and returns conditioned on observations, and to bias sampling towards actions with high returns. The paper evaluates using implicit models against state of the art offline reinforcement learning methods on the ADROIT, Mujoco, and FrankaKitchen settings.

**Summary Of The Review:**

I think this is an interesting direction, but due to the limitations in the performance of the method and drawbacks of the method itself, I think the paper is not ready for acceptance.

---

> ### Author Response · Authors · 2022-11-09
> **Response to Reviewer 3Dga**
>
> We want to thank the reviewer for taking the time to review our manuscript.
>
> > It's not clear why IRvS is better than the naive way of incorporating return information by combining IBC with RWR.
>
> IBC w/ RWR as implemented by Florence et al. throws away the bottom 50\% of the data. IRvS uses the whole dataset.
>
> >  Furthermore, IRvS only does slightly better than IBC w/ RWR, and the performance is similar on most tasks.
>
> IRvS greatly outperforms IBC w/ RWR on a myriad of tasks and performs similarly on the other ones. Thus IRvS is as good or better than IBC w/ RWR. We provide below some key highlights across a subset of our experiments in Table 1 where our method does much better than IBC w/ RWR.
>
> || IRvS $\eta^{-1}=1$ | IBC w/ RWR |
> |---| --- | --- |
> |**adroit cloned door**| 6.3 | 0.1 |
> |**adroit cloned hammer**| 3.9 | 0.3 |
> |**adroit expert hammer**| 127.2 | 118 |
> |**adroit human pen**| 84.3 | 56.7 |
> |**kitchen mixed**| 67.9 | 52.1 |
>
> >  IRvS with the best eta does perform better, but requires tuning for eta. Much more thorough analysis is needed here, as this is the main contribution of the paper.
>
> We would like to draw your attention to the 24 figures showing experiments providing a thorough analysis of the effect of eta in the Appendix.
>
> > Training EBMs with Langevin Dynamics can be difficult and requires many hyperparameters.
>
> The hyper-parameters we used here for the Langevin dynamics are precisely the same as the ones used by Florence et al.
>
> > Sometimes reward information is unavailable or difficult to get, particularly in real robotics, and the method may struggle in these cases.
>
> In the situation where no reward information is available, the method would be equivalent to IBC which is known to work well for tasks without reward information, see Florence et al.
>
> We believe our work is of significant interest to the community and hope that our clarifications above might motivate you to increase your score. If you have any other questions we are happy to provide further responses. We are considering withdrawing the paper, so it would be helpful if you could indicate your openness to increasing your score.

---

### Official Review · Reviewer_aCoh · 2022-10-25

**Confidence:** 4
**Correctness:** 3
**Technical Novelty And Significance:** 2
**Empirical Novelty And Significance:** 2
**Recommendation:** 3

**Clarity, Quality, Novelty And Reproducibility:**

Clarity:
This paper has several typos as follows.
1. $\eta^{-1} = 5.0$ in the caption of Figure 2(b) is inconsistent with the value in the sentence “In Figure 2b, we study the impact of increasing $\eta^{-1}$ to 3” on Page 4.
2. In Extrapolation of Section 5.1, “Figure 4c a)” should be “Figure 4c i).”

Novelty:
The novelty of this paper is marginally significant.

Reproducibility:
The authors provide the code for reproducibility.


**Strength And Weaknesses:**

Strength:
1. The authors leverage the exponential tilt density [1] to learn policies that can head toward the largest rewards from offline datasets collected by policies with different expertise levels.

Weaknesses:
1. The authors claim that they bridge an important gap between IBC [2] and RvS by modeling the dependencies between the state, action, and return with an implicit model on Page 6. However, noticing that IBC proposes to use the implicit model to model the dependencies between the state and action, I think the contribution of this paper is to introduce the return from RvS to the implicit model. Thus, the proposed method looks like a combination of IBC and RvS.

2. The authors conduct experiments in Section 5.1 to show the advantages of the implicit model. However, such advantages are similar to IBC, which could hurt the novelty of this paper. The authors may want to highlight the novelty of the proposed method against IBC.

3. The discussions of the empirical results in Sections 5.1 and 5.2.2 are missing. The authors may want to explain: 1) why the RvS method fails to reach either goal and converges to the purple point in Figure 4(b); 2) why the explicit methods perform better than implicit methods on the locomotion tasks.

4. The pseudo-code of the proposed method is missing.

[1] Søren Asmussen and Peter W Glynn. Stochastic simulation: algorithms and analysis, volume 57. Springer, 2007.

[2] P. Florence, C. Lynch, A. Zeng, O. A. Ramirez, A. Wahid, L. Downs, A. Wong, J. Lee, I. Mordatch, and J. Tompson. Implicit behavioral cloning. In Proceedings of the 5th Conference on Robot Learning. PMLR, 2022.

**Summary Of The Paper:**

The authors propose an Implicit Reinforcement Learning via Supervised Learning (RvS) methods by leveraging the implicit model---the composition of $\arg\min$ with a general function approximator $f_\theta$ to represent the policy ($\hat{a} = \arg\min_a f_\theta (s, a)$) [1]---instead of the traditional explicit model ($a = f_\theta (s)$) in RvS methods to solve offline RL problems. Moreover, the authors provide empirical results to show the superiority of the implicit model. Experiments demonstrate the performance improvement of the implicit model in RvS methods for offline RL.

[1] P. Florence, C. Lynch, A. Zeng, O. A. Ramirez, A. Wahid, L. Downs, A. Wong, J. Lee, I. Mordatch, and J. Tompson. Implicit behavioral cloning. In Proceedings of the 5th Conference on Robot Learning. PMLR, 2022.

**Summary Of The Review:**

I think the proposed method is a simple combination of IBC [1] and RvS. The claimed major novelty of this paper is the same as that proposed by IBC.

[1] P. Florence, C. Lynch, A. Zeng, O. A. Ramirez, A. Wahid, L. Downs, A. Wong, J. Lee, I. Mordatch, and J. Tompson. Implicit behavioral cloning. In Proceedings of the 5th Conference on Robot Learning. PMLR, 2022.

---

> ### Author Response · Authors · 2022-11-09
> **Response to Reviewer aCoh**
>
> We want to thank the reviewer for taking the time to review our manuscript.
>
> > The authors claim that they bridge an important gap between IBC [2] and RvS by modeling the dependencies between the state, action, and return with an implicit model on Page 6. However, noticing that IBC proposes to use the implicit model to model the dependencies between the state and action, I think the contribution of this paper is to introduce the return from RvS to the implicit model. Thus, the proposed method looks like a combination of IBC and RvS.
>
> || Explicit | Implicit |
> |---| --- | --- |
> |**Return independent policies**| Behavior Cloning | Implicit Behavior Cloning |
> |**Return conditional policies**| Reinforcement Learning via Supervised Learning  | Implicit Reinforcement Learning via Supervised Learning |
>
> Indeed, the proposed method is a novel way to combine implicit methods and return information. Given the promises of implicit methods in robotics (see Florence et al.), we believe that investigating this novel class of policies is a worthwhile addition to the machine learning literature. The types of Implicit models (EBMs) that we explore here are precisely what (in parallel work) LeCun, (2022) has advocated for in his position paper on 'A Path Towards Autonomous Machine Intelligence'. Making this work highly relevant to current debates in the community.
>
>
> > The authors conduct experiments in Section 5.1 to show the advantages of the implicit model. However, such advantages are similar to IBC, which could hurt the novelty of this paper. The authors may want to highlight the novelty of the proposed method against IBC.
>
> As mentioned above, IBC does not incorporate return information and thus cannot control its behavior to maximize return as it is done by IRvS.
>
>
> > The discussions of the empirical results in Sections 5.1 and 5.2.2 are missing. The authors may want to explain: 1) why the RvS method fails to reach either goal and converges to the purple point
>
> To help explain this point we propose to change this sentence: ``This represents a challenge for explicit models as they can hardly fit discontinuous functions and instead fit intermediate values``
>
> to: ``Explicit methods are known to have difficulties fitting discontinuous function (Lecun et al., 2006; Florence et al., 2021) and instead fit intermediate values between the discontinuities. We observe this phenomenon in Figure 4 c), where RvS reaches the purple point which is between the two goals and is potentially highly problematic for certain types of tasks. Reaching goal 1 would require angle $\pi/2$ and reaching goal 2 would require angle $-\pi/2$, thus the explicit model would simply output angle 0 since this is the action that would minimize the mean squared error. While the implicit methods would have equal mass on $\pi/2$ and $-pi/2$, but not on 0, thus would reach each goal with equal probability. ``
>
> Would this change make the explanation clearer?
>
> > in Figure 4(b); 2) why the explicit methods perform better than implicit methods on the locomotion tasks.
>
> Implicit methods are better than explicit models to capture multi-modality. Locomotion tasks do not have these multi modalities and thus implicit methods have no advantage.
>
> We believe that the contribution of our submission warrants a score that is higher than 3. We are considering withdrawing our submission because with one score being a 3 it becomes very unlikely for our paper to be accepted. Would it be possible for you to rapidly and clearly express your openness to increasing your score based on our clarifications?

---

> > ### Comment · Reviewer_aCoh · 2022-11-18
> > **Thanks for the authors' rebuttal.**
> >
> > Thanks for the authors' response. However, my major concerns 1 and 2 have not been properly addressed.
> >
> > 1. (Concern 1) The authors claimed that it was worthwhile to investigate this novel class (implicit models) of policies in machine learning. However, IBC has investigated the use of implicit models in offline RL. Therefore, the authors' response didn't explain why the proposed combination of IBC and RvS is novel.
> >
> > 2. (Concern 2) The authors claimed that their novelty against IBC was that they incorporated the RvS methods, which leads to Concern 1 about whether the combination method was nontrivial.
> >
> > Moreover, I have some questions about Concern 1 as follows.
> > 1. What is the important gap that the authors bridge in their combination? Why is the gap important?
> > 2. Could the authors propose some novel insights of their combination to enhance the contribution?

---

### Decision · Program_Chairs · 2023-01-20

**Decision:**

Reject

**Justification For Why Not Higher Score:**

None of the reviewers is arguing for acceptance due to the weaknesses outlined above.

**Justification For Why Not Lower Score:**

N/A

**Metareview: Summary, Strengths And Weaknesses:**

(a) Summary
The paper combines implicit behavioral cloning with Reinforcement Learning via Supervised Learning. The method is evaluated both on toy benchmarks and (simulated) robotics benchmarks and compared to baselines.

(b) Strengths
- Well written paper on an interesting topic
- Elegant method
- Good variety of experimental domains
- Impressive results on adroit and kitchen

(c) Weaknesses
- The novelty of the proposed method and its significance still remains unclear after the rebuttal
- The promised results with more seeds were not delivered. The statistical significance of the results is still questionable, as are the improvements on MuJoCo

**Summary Of Ac-Reviewer Meeting:**

N/A